# Adversarially Robust Generalization Just Requires More Unlabeled Data

## Abstract

Neural network robustness has recently been highlighted by the existence of adversarial examples. Many previous works show that the learned networks do not perform well on perturbed test data, and significantly more labeled data is required to achieve adversarially robust generalization. In this paper, we theoretically and empirically show that with just more *unlabeled data*, we can learn a model with better adversarially robust generalization. The key insight of our results is based on a risk decomposition theorem, in which the expected robust risk is separated into two parts: the *stability* part which measures the prediction stability in the presence of perturbations, and the *accuracy* part which evaluates the standard classification accuracy. As the *stability* part does not depend on any label information, we can optimize this part using unlabeled data. We further prove that for a specific Gaussian mixture problem illustrated by Schmidt et al. (2018), adversarially robust generalization can be *almost as easy as* the standard generalization in supervised learning if a sufficiently large amount of unlabeled data is provided. Inspired by the theoretical findings, we further show that a practical adversarial training algorithm that leverages unlabeled data can improve adversarial robust generalization on MNIST and Cifar-10.

## 1 Introduction

Deep learning (LeCun et al., 2015), especially deep Convolutional Neural Network (CNN) (LeCun et al., 1998), has led to state-of-the-art results spanning many machine learning fields, such as image classification (Simonyan & Zisserman, 2014; He et al., 2016; Huang et al., 2017; Hu et al., 2017), object detection (Ren et al., 2015; Redmon et al., 2016; Lin et al., 2018), semantic segmentation (Long et al., 2015; Zhao et al., 2017; Chen et al., 2018) and action recognition (Tran et al., 2015; Wang et al., 2016; 2018).

Despite the great success in numerous applications, recent studies show that deep CNNs are vulnerable to some well-designed input samples named as Adversarial Examples (Szegedy et al., 2013; Biggio et al., 2013). Take image classification as an example, for almost every commonly used well-performed CNN, attackers are able to construct a small perturbation on an input image. The perturbation is almost imperceptible to humans but can fool the model to make a wrong prediction. The problem is serious as some designed adversarial examples can be transferred among different kinds of CNN architectures (Papernot et al., 2016), which makes it possible to perform black-box attack: an attacker has no access to the model parameters or even architecture, but can still easily fool a machine learning system.

There is a rapidly growing body of work on studying how to obtain a robust neural network model. Most of the successful methods are based on adversarial training (Szegedy et al., 2013; Madry et al., 2017; Goodfellow et al., 2015; Huang et al., 2015). The high-level idea of these works is that during training, we predict the strongest perturbation to each sample against the current model and use the perturbed sample together with the correct label for gradient descent optimization. However, the learned model tends to overfit on the training data and fails to keep robust on unseen testing data. For example, using the state-of-the-art adversarial robust training method (Madry et al., 2017), the defense success rate of the learned model on the testing data is below 60% while that on the training data is almost 100%, which indicates that the robustness fails to generalize. Some theoretical results further show that it is challenging to achieve adversarially robust generalization. Fawzi et al.

(2018) proves that adversarial examples exist for any classifiers and can be transferred across different models, making it impossible to design network architectures free from adversarial attacks. Schmidt et al. (2018) shows that adversarially robust generalization requires much more labeled data than standard generalization in certain cases. Tsipras et al. (2019) presents an inherent trade-off between accuracy and robust accuracy and argues that the phenomenon comes from the fact that robust classifiers learn different features. Therefore it is hard to reach high robustness for standard training methods.

Given the challenge of the task and previous findings, in this paper, we provide several theoretical and empirical results towards better adversarially robust generalization. In particular, we show that we can learn an adversarially robust model which generalizes well if we have plenty of *unlabeled data*, and the labeled sample complexity for adversarially robust generalization in Schmidt et al. (2018) can be largely reduced if unlabeled data is used. First, we show that the expected robust risk can be upper bounded by the sum of two terms: a stability term which measures whether the model can output consistent predictions under perturbations, and an accuracy term which evaluates whether the model can make correct predictions on natural samples. Given the stability term does not rely on ground truth labels, unlabeled data can be used to minimize this term and thus improve the generalization ability. Second, we prove that for the Gaussian mixture problem defined in Schmidt et al. (2018), if unlabeled data can be used, adversarially robust generalization will be almost as easy as the standard generalization in supervised learning (*i.e.* using the same number of labeled samples under similar conditions). Inspired by the theoretical findings, we provide a practical algorithm that can learn from both labeled and unlabeled data for better adversarially robust generalization. Our experiments on MNIST and Cifar-10 show that the method achieves better performance, which verifies our theoretical findings.

Our contributions are in three folds.

- In Section 3.2.1, we provide a theorem to show that unlabeled data can be naturally used to improve the expected robust risk in general setting and thus leveraging unlabeled data is a way to improve adversarially robust generalization.

- In Section 3.2.2, we discuss a specific Gaussian mixture problem introduced in Schmidt et al. (2018). In Schmidt et al. (2018), the authors proved that in this case, the labeled sample complexity for robust generalization is significantly larger than that for standard generalization. As an extension of this work, we prove that in this case, the labeled sample complexity for robust generalization can be the same as that for standard generalization if we have enough unlabeled data.

- Inspired by our theoretical findings, we provide an adversarial robust training algorithm using both labeled and unlabeled data. Our experimental results show that the algorithm achieves better performance than baseline algorithms on MNIST and Cifar-10, which empirically proves that unlabeled data can help improve adversarially robust generalization.

## 2 RELATED WORKS

**Adversarial attacks and defense**  Most previous works study how to attack a neural network model using small perturbations under certain norm constraints, such as $l_\infty$ norm or $l_2$ norm. For the $l_\infty$ constraint, Fast Gradient Sign Method (FGSM) (Goodfellow et al., 2015) finds a direction to which the perturbation increases the classification loss at an input point to the greatest extent; Projected Gradient Descent (PGD) (Madry et al., 2017) extends FGSM by updating the direction of the attack in an iterative manner and clipping the modifications in the norm range after each iteration. For the $l_2$ constraint, DeepFool (Moosavi-Dezfooli et al., 2016) iteratively computes the minimal norm of an adversarial perturbation by linearizing around the input in each iteration. C&W attack (Carlini & Wagner, 2016) is a comprehensive approach that works under both norm constraints. In this work, we focus on learning a robust model to defend the white-box attack, *i.e.* the attacker knows the model parameters and thus can use the algorithms above to attack the model.

There are a large number of papers about defending against adversarial attacks, but the result is far from satisfactory. Remarkably, Athalye et al. (2018) shows most defense methods take advantage of so-called "gradient mask" and provides an attacking method called BPDA to correct the gradients. So far, adversarial training (Madry et al., 2017) has been the most successful white-box defense al-

gorithm. By modeling the learning problem as a mini-max game between the attacker and defender, the robust model can be trained using iterative optimization methods. Some recent papers (Wang et al., 2019; Gao et al., 2019) theoretically prove the convergence of adversarial training. Moreover, Shafahi et al. (2019); Zhang et al. (2019a) propose ways to accelerate the speed of adversarial training. Adversarial logit pairing (Kannan et al., 2018) and TRADES (Zhang et al., 2019b) further improve adversarial training by decomposing the prediction error as the sum of classification error and boundary error, and Wang et al. (2019) proposes to improve adversarial training by evaluating the quality of adversarial examples using the FOSC metric.

**Semi-supervised learning**   Using unlabeled data to help the learning process has been proved promising in different applications (Rasmus et al., 2015; Zhang & Shi, 2011; Elworthy, 1994). Many approaches use regularizers called "soft constraints" to make the model "behave" well on unlabeled data. For example, transductive SVM (Joachims, 1999) uses prediction confidence as a soft constraint, and graph-based SSL (Belkin et al., 2006; Talukdar & Crammer, 2009) requires the model to have similar outputs at endpoints of an edge. The most related work to ours is the consistency-based SSL. It uses *consistency* as a soft constraint, which encourages the model to make consistent predictions on unlabeled data when a small perturbation is added. The consistency metric can be either computed by the model's own predictions, such as the Π model (Sajjadi et al., 2016), Temporal Ensembling (Laine & Aila, 2016) and Virtual Adversarial Training (Miyato et al., 2018), or by the predictions of a teacher model, such as the mean teacher model (Tarvainen & Valpola, 2017).

**Semi-supervised learning for adversarially robust generalization**   There are three other concurrent and independent works (Carmon et al., 2019; Uesato et al., 2019; Najafi et al., 2019) which also explore how to use unlabeled data to help adversarially robust generalization. We describe the three works below, and compare them with ours. See also Carmon et al. (2019) and Uesato et al. (2019) for the comparison of all the four works from their perspective.

Najafi et al. (2019) investigate the robust semi-supervised learning from the distributionally robust optimization perspective. They assign soft labels to the unlabeled data according to an adversarial loss and train such images together with the labeled ones. Results on a wide range of tasks show that the proposed algorithm improves the adversarially robust generalization. Both Najafi et al. (2019) and we conduct semi-supervised experiments by removing labels from the training data.

Uesato et al. (2019) study the Gaussian mixture model of Schmidt et al. (2018) and theoretically show that a self-training algorithm can successfully leverage unlabeled data to improve adversarial robustness. They extend the self-training algorithm to the real image dataset Cifar10, augment it with unlabeled Tiny Image dataset and improve state-of-the-art adversarial robustness. They show strong improvements in the low labeled data regimes by removing most labels from CIFAR-10 and SVHN. In our work, we also study the Gaussian mixture model and show that a slightly different algorithm can improve adversarially robust generalization as well. We observe similar improvements using our algorithm on Cifar-10 and MNIST.

Carmon et al. (2019) obtain similar theoretical and empirical results as in Uesato et al. (2019), and offer a more comprehensive analysis of other aspects. They show that by using unlabeled data and robust self-training, the learned models can obtain better certified robustness against all possible attacks. Moreover, they study the impact of different training components on the final model performance, such as the size of unlabeled data. We also study the influence of different factors in our experiments and have similar observations.

## 3   MAIN RESULTS

In this section, we illustrate the benefits of using unlabeled data for robust generalization from a theoretical perspective.

### 3.1   NOTATIONS AND DEFINITIONS

We consider a standard classification task with an underlying data distribution $\mathcal{P}_{XY}$ over pairs of examples $x \in \mathbb{R}^d$ and corresponding labels $y \in \{1, 2, \cdots, K\}$. Usually $\mathcal{P}_{XY}$ is unknown and we can only access to $S = \{(x_1, y_1), \cdots, (x_n, y_n)\}$ in which $(x_i, y_i)$ is independent and identically

drawn from $\mathcal{P}_{XY}$, $i = 1, 2, \cdots, n$. For ease of reference, we denote this empirical distribution as $\hat{\mathcal{P}}_{XY}$ (*i.e.* the uniform distribution over *i.i.d.* sampled data). We also assume that we are given a suitable loss function $l(f(x), y)$, where $f \in \mathcal{F}$ is parameterized by $\theta$. The standard loss function is the zero-one loss, *i.e.* $l^{0/1}(y', y) = \mathbb{I}[y' \neq y]$. Due to its discontinuous and non-differentiable nature, surrogate loss functions such as cross-entropy or mean square loss are commonly used during optimization.

Our goal is to find an $f \in \mathcal{F}$ that minimizes the expected classification risk. Without loss of any generality, our theory is mainly based on the *binary classification* problem, *i.e.* $K = 2$. All theorems below can be easily extended to the multi-class classification problem. For a binary classification problem, the expected classification risk is defined as below.

**Definition 1.** (Expected classification risk). Let $\mathcal{P}_{XY}$ be a probability distribution over $\mathbb{R}^d \times \{\pm 1\}$. The expected classification risk $R$ of a classifier $f : \mathbb{R}^d \to \{-1, 1\}$ under distribution $\mathcal{P}_{XY}$ and loss function $l$ is defined as

$$R = \mathbb{E}_{(x,y) \sim \mathcal{P}_{XY}} l(f(x), y) \tag{1}$$

We use $R(f)$ to denote the classification risk under the underlying distribution and use $\hat{R}(f)$ to denote the classification risk under the empirical distribution. We use $R^{0/1}(f)$ to denote the risk with the zero-one loss function. The classification risk characterizes whether the model $f$ is accurate. However, we also care about whether $f$ is *robust*. For example, when input $x$ is an image, we hope a small change (perturbation) to $x$ will not change the prediction of $f$. To this end, Schmidt et al. (2018) defines *expected robust classification risk* as follows.

**Definition 2.** (Expected robust classification risk). Let $\mathcal{P}_{XY}$ be a probability distribution over $\mathbb{R}^d \times \{\pm 1\}$ and $\mathcal{B} : \mathbb{R}^d \to \mathscr{P}\left(\mathbb{R}^d\right)$ be a perturbation set. Then the $\mathcal{B}$-robust classification risk $R_{\mathcal{B}-\text{robust}}$ of a classifier $f : \mathbb{R}^d \to \{-1, 1\}$ under distribution $\mathcal{P}_{XY}$ and loss function $l$ is defined as

$$R_{\mathcal{B}-\text{robust}} = \mathbb{E}_{(x,y) \sim \mathcal{P}_{XY}} \sup_{x' \in \mathcal{B}(x)} l(f(x'), y) \tag{2}$$

Again, we use $R_{\mathcal{B}-\text{robust}}(f)$ to denote the expected robust classification risk under the underlying distribution and use $\hat{R}_{\mathcal{B}-\text{robust}}(f)$ to denote the expected robust classification risk under the empirical distribution. We use $R^{0/1}_{\mathcal{B}-\text{robust}}(f)$ to denote the robust risk with the zero-one loss function. In real practice, the most commonly used setting is the perturbation under $\epsilon$-bounded $l_\infty$ norm constraint $\mathcal{B}^{\epsilon}_{\infty}(x) = \left\{x' \in \mathbb{R}^d \mid \|x' - x\|_\infty \leq \varepsilon\right\}$. For simplicity, we refer to the robustness defined by this perturbation set as $\ell^{\epsilon}_{\infty}$-robustness.

## 3.2 ROBUST GENERALIZATION ANALYSIS

Our first result (Section 3.2.1) shows that unlabeled data can be used to improve adversarially robust generalization in general setting. Our second result (Section 3.2.2) shows that for a specific learning problem defined on Gaussian mixture model, compared to previous work (Schmidt et al., 2018), the sample complexity for robust generalization can be significantly reduced by using unlabeled data. Both results suggest that using unlabeled data is a natural way to improve adversarially robust generalization. All detailed proofs of the theorems and lemmas in this section can be found in the appendix.

### 3.2.1 GENERAL RESULTS

In this subsection, we show that the expected robust classification risk can be bounded by the sum of two terms. The first term only depends on the hypothesis space and the *unlabeled data*, and the second term is a standard PAC bound.

**Theorem 1.** *Let $\mathcal{F}$ be the hypothesis space. Let $S = (x_i, y_i)_{i=1}^{n}$ be the set of $n$ i.i.d. samples drawn from the underlying distribution $\mathcal{P}_{XY}$. For any function $f \in \mathcal{F}$, with probability at least $1 - \delta$ over the random draw of $S$, we have*

$$R^{0/1}_{\mathcal{B}-robust}(f) \leq \underbrace{\mathbb{E}_{x \sim \mathcal{P}_X} \sup_{x' \in \mathcal{B}(x)} (\mathbb{I}(f(x') \neq f(x))}_{(1)} + \underbrace{\hat{R}^{0/1}(f) + Rad_S(\mathcal{F}) + 3\sqrt{\frac{\log \frac{2}{\delta}}{2n}}}_{(2)}, \tag{3}$$

*where (1) is a term that can be optimized with only unlabeled data and (2) is the standard PAC generalization bound. $\mathcal{P}_X$ is the marginal distribution for $\mathcal{P}_{XY}$ and $Rad_S(\mathcal{F})$ is the empirical Rademacher complexity of hypothesis space $\mathcal{F}$.*

From Theorem 1, we can see that the expected robust classification risk is bounded by the sum of two terms: the first term only involves the marginal distribution $\mathcal{P}_X$ and the second term is the standard PAC generalization error bound. This shows that the expected robust risk minimization can be achieved by jointly optimizing the two terms simultaneously: we can optimize the first term using unlabeled data sampled from $P_X$ and optimize the second term using labeled data sampled from $P_{XY}$, which is the same as the standard supervised learning.

While Cullina et al. (2018) suggests that in the standard PAC learning scenario (only labeled data is considered), the generalization gap of robust risk can be sometimes uncontrollable by the capacity of hypothesis space $\mathcal{F}$, our results show that we can mitigate this problem by introducing unlabeled data. In fact, our following result shows that with enough unlabeled data, learning a robust model can be almost as easy as learning a standard model.

### 3.2.2 LEARNING FROM GAUSSIAN MIXTURE MODEL

The learning problem defined on Gaussian mixture model is illustrated in Schmidt et al. (2018) as an example to show adversarially robust generalization needs much more labeled data compared to standard generalization. In this subsection, we show that for this specific problem, just using more unlabeled data is enough to achieve adversarially robust generalization. For completeness, we first list the results in Schmidt et al. (2018) and then show our theoretical findings.

**Definition 3.** (Gaussian mixture model (Schmidt et al., 2018)). Let $\theta^* \in \mathbb{R}^d$ be the per-class mean vector and let $\sigma > 0$ be the variance parameter. Then the $(\theta^*, \sigma)$-Gaussian mixture model is defined by the following distribution $\mathcal{P}_{XY}$ over $(x, y) \in \mathbb{R}^d \times \{\pm 1\}$: First, draw a label $y \in \{\pm 1\}$ uniformly at random. Then sample the data point $x \in \mathbb{R}^d$ from $\mathcal{N}(y \cdot \theta^*, \sigma^2 \mathbf{I}_d)$.

Given the samples from the distribution defined above, the learning problem is to find a linear classifier to predict label $y$ from $x$. Schmidt et al. (2018) proved the following sample complexity bound for standard generalization.

**Theorem 2.** *(Theorem 4 in Schmidt et al. (2018)). Let $(x, y)$ be drawn from the $(\theta^\star, \sigma)$-Gaussian mixture model with $\|\theta^\star\|_2 = \sqrt{d}$ and $\sigma \leq c \cdot d^{1/4}$ where $c$ is a universal constant. Let $\hat{w} \in \mathbb{R}^d$ be the vector $\hat{w} = y \cdot x$. Then with high probability, the expected classification risk of the linear classifier $f_{\hat{w}}$ using 0-1 loss is at most 1%.*

Theorem 2 suggests that we can learn a linear classifier with low classification risk (e.g., 1%) even if there is only one labeled data. However, the following theorem shows that for adversarially robust generalization under $\ell_\infty^\epsilon$ perturbation, significantly more labeled data is required.

**Theorem 3.** *(Theorem 6 in Schmidt et al. (2018)). Let $g_n$ be any learning algorithm, i.e. a function from $n$ samples to a binary classifier $f_n$. Moreover, let $\sigma = c_1 \cdot d^{1/4}$, let $\epsilon \geq 0$, and let $\theta \in \mathbb{R}^d$ be drawn from $\mathcal{N}(0, \mathbf{I}_d)$. We also draw $n$ samples from the $(\theta, \sigma)$-Gaussian mixture model. Then the expected $\ell_\infty^\epsilon$-robust classification risk of $f_n$ using 0-1 loss is at least $\frac{1}{2}(1 - 1/d)$ if the number of labeled data $n \leq c_2 \frac{\epsilon^2 \sqrt{d}}{\log d}$.*

As we can see from above theorem, the sample complexity for robust generalization is larger than that of standard generalization by $\sqrt{d}$. This shows that for high-dimensional problems, adversarial robustness can provably require a significantly larger number of samples. We provide a new result which shows that the learned model can be robust if there is only one labeled data and sufficiently many unlabeled data. Our theorem is stated as follow:

**Theorem 4.** *Let $(x^L, y^L)$ be **a labeled point** drawn from $(\theta^*, \sigma)$-Gaussian mixture model $\mathcal{P}_{XY}$ with $\|\theta^*\|_2 = \sqrt{d}$ and $\sigma = O(d^{1/4})$. Let $x_1^U, \cdots, x_n^U$ be $n$ **unlabeled points** drawn from $\mathcal{P}_X$. Let $v \in \mathbb{R}^d$ such that $v \in \arg\max_{\|v\|=1} \sum_{i=1}^n (v^\top x_i^U)^2$. Let $\hat{w} = sign(y^L \cdot v^\top x^L)v$. Then there exists a constant $D$ such that for any $d \geq D$, with high probability, the expected $\ell_\infty^\epsilon$-robust classification risk of $f_{\hat{w}}$ using 0-1 loss is at most 1% when **the number of unlabeled points** $n = \Omega(d)$ and $\epsilon \leq \frac{1}{2}$.*

From Theorem 4, we can see that when the number of unlabeled points is significant, we can learn a highly accurate and robust model using only one labeled point.

**Proof sketch** The learning process can be intuitively described as the following three steps: in the first step, we use unlabeled data to estimate the *direction* of $\theta^*$ although we do not know the label that $\theta^*$ (or $-\theta^*$) corresponds to. Specifically, we choose the direction $v$ which maximizes the quantity $\sum_{i=1}^n (v^\top x_i^U)^2$ which can be viewed as a measure of the confidence at data points. In the second step, we use the given labeled point to determine the *sign* of $\theta^*$ with high probability, we note that when the direction is correctly estimated in the first step, then the only one labeled point is sufficient to give the correct sign with high probability. Finally, we give a good estimation of $\theta^*$ by combining the two steps above and learn a robust classifier. The three key lemmas corresponding to the three steps are listed below ($c_i$ are constants for $i = 0, 1, 2, 3$).

**Lemma 1.** *Under the same setting as Theorem 4, suppose that $n > d$ and $\sigma\sqrt{\frac{\sigma^2+d}{nd}} < \frac{1}{128}$. Then, with probability at least $1 - c_1 e^{-c_2 n \min\{\frac{\sqrt{d}c_3}{\sigma}, (\frac{\sqrt{d}c_3}{\sigma})^2\}}$, there is a unique unit maximal eigenvector $v$ of the sample covariance matrix $\hat{\Sigma} = \frac{1}{n}\sum_{i=1}^n x_i^U x_i^{U\top}$ such that*

$$\left\| v - \frac{\theta^*}{\sqrt{d}} \right\|_2 \leq c_0 \sigma \sqrt{\frac{\sigma^2 + d}{nd}} + c_3 \tag{4}$$

**Lemma 2.** *Under the same setting as Theorem 4, suppose $v$ is a unit vector such that $\left\| v - \frac{\theta^*}{\sqrt{d}} \right\|_2 \leq \tau$ for some constant $\tau < \sqrt{2}$. Then with probability at least $1 - \exp(-\frac{d(1-\frac{\tau^2}{2})^2}{2\sigma^2})$, we have*

$$sign(y^L \cdot v^\top x^L) v^\top \theta^* > 0 \tag{5}$$

**Lemma 3.** *(Lemma 20 in Schmidt et al. (2018)). Under the same setting as Theorem 4, for any $p \geq 1$ and $\epsilon \geq 0$, and for any unit vector $\hat{w}$ such that $\langle \hat{w}, \theta^\star \rangle \geq \epsilon \|\hat{w}\|_p^*$, where $\|\cdot\|_p^*$ is the dual norm of $\|\cdot\|_p$, the linear classifier $f_{\hat{w}}$ has $\ell_p^\epsilon$-robust classification risk at most $\exp\left(-\frac{(\langle \hat{w}, \theta^*\rangle - \epsilon\|\hat{w}\|_p^*)^2}{2\sigma^2}\right)$.*

Our theoretical findings suggest that we can improve the adversarially robust generalization using unlabeled data. In the next section, we will present a practical algorithm for real applications, which further verifies our main results.

---

**Algorithm 1** Generalized Virtual Adversarial Training over labeled and unlabeled data

---

1: **Input**: Datasets $S^L$ and $S^U$. Hypothesis space $\mathcal{F}$. Coefficient $\lambda$. PGD step size $\delta$. Number of PGD steps $k$. Maximum $l_\infty$ norm of perturbation $\epsilon$.
2: **for** each iteration **do**
3:     Sample a mini-batch of labeled data $\hat{S}^L$ from $S^L$.
4:     Sample a mini-batch of unlabeled data $\hat{S}^U$ from $S^U$.
5:     **for** each $x \in \hat{S}^L \cup \hat{S}^U$ **do**
6:         Fix $f$ and attack $x$ with PGD-$(k, \epsilon, \delta)$ on loss $L_1/L_2$ to obtain $x'$.
7:         Perform gradient descent on $f$ over the perturbed samples on loss $L^{SSL}$.
8:     **end for**
9: **end for**

---

## 4   Algorithm and experiments

### 4.1   Practical algorithm

Let $S^L = \{(x_1^L, y_1^L), \cdots, (x_n^L, y_n^L)\}$ be a set of labeled data and $S^U = \{x_1^U, \cdots, x_m^U\}$ be a set of unlabeled data. Motivated by the theory in the previous section, to achieve better adversarially robust generalization, we can optimize the classifier to be accurate on $S^L$ and robust on $S^L \cup S^U$. This is also equivalent to making the classifier accurate and robust on $S^L$ and robust on $S^U$. Therefore, we design two loss terms on $S^L$ and $S^U$ separately.

For the labeled dataset $S^L$, we use the standard $\ell_\infty^\epsilon$-robust adversarial training objective function, i.e.,

$$L_1(f, S^L) = \frac{1}{n}\sum_{i=1}^n \max_{x_i' \in \mathcal{B}_\infty^\epsilon(x_i)} l^{CE}(f(x_i'), y_i). \tag{6}$$

Following the most common setting, during training, the classifier outputs a probability distribution over categories and is evaluated by cross-entropy loss defined as

$$l^{CE}(f(x), y) = -\sum_{k=1}^{K} \log f_k(x)\mathbb{I}[y = k] \tag{7}$$

where $f_k(x)$ is the output probability for category $k$.

For unlabeled data $S^U$, we use an objective function which measures robustness without ground truth

$$L_2(f, S^U) = \frac{1}{m}\sum_{i=1}^{m} \max_{x_i' \in \mathcal{B}_\infty^\epsilon(x_i)} l^{CE}(f(x_i'), \hat{y}_i), \text{where } \hat{y}_i = \arg\max_k\{f_k(x_i)\}. \tag{8}$$

Putting the two objective functions together, our training loss is defined as a combination of $L_1$ and $L_2$ as follows:

$$L^{SSL}(f, S^L, S^U) = L_1(f, S^L) + \lambda L_2(f, S^U). \tag{9}$$

Here $\lambda > 0$ is a coefficient to trade off the two loss terms. In real practice, we use iterative optimization methods to learn the function $f$. In the inner loop, we fix the model and use Projected Gradient Descent (PGD) to learn the attack $x'$ for any $x$. In the outer loop, we use stochastic gradient descent to optimize $f$ on the perturbed $x'$s. The general training process is shown in Algorithm 1.

**Remark**  We notice that Algorithm 1 is a generalized version of Virtual Adversarial Training (VAT) (Miyato et al., 2018). When setting the PGD step $k = 1$, the algorithm is almost equivalent to the original VAT algorithm, which is particular useful for improving standard generalization. However, according to our experimental results below, setting $k = 1$ does not help improve adversarial robust generalization. The improvement of adversarial robust generalization using unlabeled data exists when setting a relatively larger $k$.

## 4.2   EXPERIMENTAL SETTING

We verify Algorithm 1 on MNIST and Cifar-10. Following Madry et al. (2017), we use the Resnet model and modify the network incorporating wider layers by a factor of 10. This results in a network with five residual units with (16, 160, 320, 640) filters each. During training, we apply data augmentation including random crops and flips, as well as per image standardization. The initial learning rate is 0.1, and decay by a factor of 10 twice during training. In the inner loop, we run a 7-step PGD with step size $\frac{2}{255}$ for each mini-batch. The perturbation is constrained to be $\frac{8}{255}$ under $l_\infty$ norm.

Following many previous works (Laine & Aila, 2016; Tarvainen & Valpola, 2017; Miyato et al., 2018; Athiwaratkun et al., 2019), we sample $5k/10k$ labeled data from the training set and use them as labeled data. We mask out the labels of the remaining images in the training set and use them as unlabeled data. By doing this, we conduct two semi-supervised learning tasks and call them the $5k/10k$ experiments. In a mini-batch, we sample 25/50 labeled images and 225/200 unlabeled images for the $5k/10k$ experiment respectively. In both experiments, we use several different values of $\lambda$ as an ablation study for this hyperparameter by setting $\lambda = 0.1, 0.2, 0.3$. Learning rate is decayed at the $60^{th}$ and the $120^{th}$ epoch. We use the original PGD-based adversarial training (Madry et al., 2017) on the sampled $5k/10k$ labeled data as the baseline algorithm for comparison (referred to as PGD-adv). Our algorithm is referred to as Ours.

## 4.3   EXPERIMENTAL RESULTS

We list all results of the $5k/10k$ experiments in Tables 1 and 2. We use five criteria to evaluate the performance of the model: the natural training/test accuracy ($NA_{train}$ and $NA_{test}$), the robust training/test accuracy using PGD-7 attack ($RA_{train}$ and $RA_{test}$) and the defense success rate (DSR).

First, we can see that in both experiments, the robust test accuracy is improved when we use unlabeled data. For example, on Cifar-10 the robust test accuracy of the models trained under SSL with $\lambda = 0.3$ for the $5k/10k$ experiments increase by 3.0/5.0 percents compared to the PGD-adv baselines. We also check the defense success rate which evaluates whether the model is robust given

Table 1: SSL experiment with $5k/10k$ labeled points on MNIST (%)

|  |  | $\text{NA}_{train}$ | $\text{NA}_{test}$ | $\text{RA}_{train}$ | $\text{RA}_{test}$ | DSR |
|---|---|---|---|---|---|---|
|  | PGD-adv on $5k$ | 98.31 | 98.38 | 96.95 | 96.89 | 98.49 |
| $5k$ | Ours ($k = 7, \lambda = 0.1$) | 98.36 | 98.54 | 97.82 | 97.19 | 98.63 |
|  | Ours ($k = 7, \lambda = 0.2$) | 98.43 | 98.55 | 98.18 | 97.28 | 98.71 |
|  | Ours ($k = 7, \lambda = 0.3$) | 98.56 | 98.56 | 98.46 | 97.31 | 98.73 |
|  | PGD-adv on $10k$ | 98.91 | 98.83 | 97.96 | 97.64 | 98.80 |
| $10k$ | Ours ($k = 7, \lambda = 0.1$) | 98.92 | 98.92 | 98.55 | 97.91 | 98.98 |
|  | Ours ($k = 7, \lambda = 0.2$) | 98.90 | 98.89 | 98.76 | 97.93 | 99.03 |
|  | Ours ($k = 7, \lambda = 0.3$) | 98.93 | 98.87 | 98.77 | 98.01 | 99.13 |
|  | PGD-adv on $50k$ | 99.89 | 99.44 | 99.77 | 98.84 | 99.40 |

Table 2: SSL experiment with $5k/10k$ labeled points on Cifar-10 (%)

|  |  | $\text{NA}_{train}$ | $\text{NA}_{test}$ | $\text{RA}_{train}$ | $\text{RA}_{test}$ | DSR |
|---|---|---|---|---|---|---|
|  | PGD-adv on $5k$ | 61.18 | 60.57 | 32.40 | 30.54 | 50.42 |
| $5k$ | Ours ($k = 7, \lambda = 0.1$) | 63.24 | 60.44 | 32.97 | 30.90 | 51.13 |
|  | Ours ($k = 7, \lambda = 0.2$) | 61.73 | 60.71 | 35.20 | 32.96 | 54.29 |
|  | Ours ($k = 7, \lambda = 0.3$) | 61.88 | 60.46 | 35.07 | 33.54 | 55.47 |
|  | Ours ($k = 1, \lambda = 0.3$) | 68.15 | 67.14 | 0.13 | 0.12 | 0.00 |
|  | PGD-adv on $10k$ | 78.80 | 73.79 | 45.60 | 37.48 | 50.79 |
| $10k$ | Ours ($k = 7, \lambda = 0.1$) | 78.24 | 72.92 | 47.96 | 38.86 | 53.29 |
|  | Ours ($k = 7, \lambda = 0.2$) | 78.74 | 73.16 | 51.20 | 41.18 | 56.29 |
|  | Ours ($k = 7, \lambda = 0.3$) | 78.95 | 73.35 | 52.24 | 42.48 | 57.91 |
|  | Ours ($k = 1, \lambda = 0.3$) | 81.43 | 78.64 | 2.22 | 2.27 | 0.03 |
|  | PGD-adv on $50k$ | 99.91 | 85.40 | 96.71 | 49.99 | 58.54 |

the prediction is correct. As we can see from the last column in Tables 1 and 2, the defense success rate of models trained using our proposed method is much higher than the baselines. In particular, the defense success rate of the model trained with $\lambda = 0.3$ in the $10k$ experiment is competitive to the model trained using PGD-adv on the whole dataset. This clearly shows the advantage of our proposed algorithm.

Second, we can also see the influence of the value of $\lambda$. The model trained with a larger $\lambda$ has higher robust accuracy. For example, in the $10k$ experiment, the robust test accuracy of the model trained with $\lambda = 0.3$ is more than $3\%$ better than that with $\lambda = 0.1$. However, we observe that training will become hard to converge if $\lambda > 0.5$.

Third, using larger $k$ produces more robust models. As we can see from the table, in the $5k/10k$ experiment, relatively higher natural training/test accuracy can be achieved by setting $k = 1$ (vanilla VAT algorithm). However, the robust training/testing accuracy are significantly worse and are near zero. This clearly shows that using a stronger attack on both labeled and unlabeled data leads to better adversarially robust generalization, which is also consistent with our theory.

## 5 CONCLUSION

In this paper, we theoretically and empirically show that with just more unlabeled data, we can learn models with better adversarially robust generalization. We first give an expected robust risk decomposition theorem and then show that for a specific learning problem on the Gaussian mixture model, the adversarially robust generalization can be almost as easy as standard generalization. Based on these theoretical results, we develop an algorithm which leverages unlabeled data during training and empirically show its advantage. As future work, we will study the sample complexity of unlabeled data for broader function classes and solve more challenging real tasks.

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

## A    BACKGROUND ON GENERALIZATION AND RADEMACHER COMPLEXITY

The Rademacher complexity is a commonly used capacity measure for a hypothesis space.

**Definition 4.** Given a set $S = \{x_1, ..., x_n\}$ of $n$ samples, the empirical Rademacher complexity of function class $\mathcal{F}$ (mapping from $\mathbb{R}^d$ to $\mathbb{R}$) is defined as:

$$Rad_S(\mathcal{F}) = \frac{1}{n} \mathbb{E}_{\epsilon} \left[ \sup_{f \in \mathcal{F}} \sum_{i=1}^n \epsilon_i f(x_i) \right], \tag{10}$$

where $\epsilon = (\epsilon_1, \cdots, \epsilon_n)^\top$ contains i.i.d. random variables drawn from the Rademacher distribution unif($\{1, -1\}$).

By using the Rademacher complexity, we can directly provide an upper bound on the generalization error.

**Theorem 5.** *(Theorem 3.5 in Mohri et al. (2012)). Suppose $l(\cdot, \cdot)$ is the $0-1$ loss, let $S = (x_i, y_i)_{i=1}^n$ be the set of $n$ i.i.d. samples drawn from the underlining distribution $\mathcal{P}_{XY}$. Let $\mathcal{F}$ be the hypothesis space, then with probability at least $1 - \delta$ over $S$, for any $f \in \mathcal{F}$:*

$$R(f) \leq \hat{R}(f) + Rad_S(\mathcal{F}) + 3\sqrt{\frac{\log \frac{2}{\delta}}{2n}} \tag{11}$$

## B    PROOF OF THEOREM 1

*Proof.* For indicator function $\mathbb{I}$, we have for any $x$,

$$\mathbb{I}(f(x') \neq y) \leq \mathbb{I}(f(x) \neq y) + \mathbb{I}(f(x) \neq f(x')). \tag{12}$$

According to Definition 2, we have

$$\begin{aligned}
R_{\mathcal{B}-\text{robust}}(f) &= \mathbb{E}_{(x,y) \sim \mathcal{P}_{XY}} \sup_{x' \in \mathcal{B}(x)} l(f(x'), y) = \mathbb{E}_{(x,y) \sim \mathcal{P}_{XY}} \sup_{x' \in \mathcal{B}(x)} \mathbb{I}(f(x') \neq y) \\
&\leq \mathbb{E}_{(x,y) \sim \mathcal{P}_{XY}} \sup_{x' \in \mathcal{B}(x)} (\mathbb{I}(f(x) \neq y) + \mathbb{I}(f(x) \neq f(x'))) \\
&= \mathbb{E}_{x \sim \mathcal{P}_X} \sup_{x' \in \mathcal{B}(x)} (\mathbb{I}(f(x') \neq f(x)) + \mathbb{E}_{(x,y) \sim \mathcal{P}_{XY}} l(f(x), y) \\
&= \mathbb{E}_{x \sim \mathcal{P}_X} \sup_{x' \in \mathcal{B}(x)} (\mathbb{I}(f(x') \neq f(x)) + R(f),
\end{aligned} \tag{13}$$

where (13) is derived from (12). We further use Theorem 5 to bound $R(f)$. It is easy to verify that with probability at least $1 - \delta$, for any $f \in \mathcal{F}$:

$$R_{\mathcal{B}-\text{robust}}(f) \leq \mathbb{E}_{x \sim \mathcal{P}_X} \sup_{x' \in \mathcal{B}(x)} (\mathbb{I}(f(x') \neq f(x)) + R(f)$$

$$\leq \mathbb{E}_{x \sim \mathcal{P}_X} \sup_{x' \in \mathcal{B}(x)} (\mathbb{I}(f(x') \neq f(x)) + \hat{R}(f) + Rad_S(\mathcal{F}) + 3\sqrt{\frac{\log \frac{2}{\delta}}{2n}}$$

which completes the proof.    $\square$

## C    PROOF OF THEOREM 4

For convenience, in this section, we use $c_i$ or $c_i'$ to denote some *universal constants*, where $i = 0, 1, 2, 3, 4$.

In the proof of Theorem 4, we will use the concentration bound for covariance estimation in Wainwright (2019). We first introduce the definition of spiked covariance ensemble.

**Definition 5.** (Spiked covariance ensemble). A sample $x_i \in \mathbb{R}^d$ from the spiked covariance ensemble takes the form

$$x_i = \sqrt{\nu} \xi_i \theta_0 + w_i, \tag{14}$$

where $\xi_i \in \mathbb{R}$ is a zero-mean random variable with unit variance, $\nu \in \mathbb{R}$ is a fixed scalar, $\theta_0 \in \mathbb{R}^d$ is a fixed unit vector and $w_i \in \mathbb{R}^d$ is a random vector independent of $\xi_i$, with zero mean and covariance matrix $\mathbf{I}_d$.

To see why spiked covariance ensemble model is useful, we note that the Gaussian mixture model is its special case. Specifically, let $x_i^U$'s be the unlabeled data in Theorem 4. Then $x_i^U$ follows the Gaussian mixture distribution $\frac{1}{2}\mathcal{N}(\theta^*, \sigma^2 \mathbf{I}_d) + \frac{1}{2}\mathcal{N}(-\theta^*, \sigma^2 \mathbf{I}_d)$, and $\frac{x_i^U}{\sigma}$ is a spiked covariance ensemble with parameter $\nu = \frac{d}{\sigma^2}$, $\xi_i$ uniformly distributed on $\{\pm 1\}$, $w_i \sim \mathcal{N}(0, \mathbf{I}_d)$ and $\theta_0 = \frac{\theta^*}{\sqrt{d}}$.

The following theorem from Wainwright (2019) characterizes the concentration property of spiked covariance ensemble, which we will further use to bound the robust classification error. Intuitively, the theorem says that we can approximately recover $\theta_0$ in the spiked covariance ensemble model using the top eigenvector $\hat{\theta}$ of the sample covariance matrix $\hat{\Sigma}$.

**Theorem 6.** *(Concentration of covariance estimation, see Corollary 8.7 in Wainwright (2019)). Given i.i.d. samples $\{x_i\}_{i=1}^n$ from the spiked covariance ensemble with sub-Gaussian tails (which means both $\xi_i$ and $w_i$ are sub-Gaussian with parameter at most one), suppose that $n > d$ and $\sqrt{\frac{\nu+1}{\nu^2}}\sqrt{\frac{d}{n}} < \frac{1}{128}$. Then, with probability at least $1 - c_1 e^{-c_2 n \min\{\sqrt{\nu}c_3, \nu c_3^2\}}$, there is a unique maximal eigenvector $\hat{\theta}$ of the sample covariance matrix $\hat{\Sigma} = \frac{1}{n}\sum_{i=1}^n x_i x_i^\top$ such that*

$$\left\|\hat{\theta} - \theta_0\right\|_2 \leq c_0 \sqrt{\frac{\nu+1}{\nu^2}}\sqrt{\frac{d}{n}} + c_3. \tag{15}$$

Using the theorem above, we can show that for the Gaussian mixture model, one of the top unit eigenvector of the sample covariance matrix is approximately $\frac{\theta^*}{\sqrt{d}}$. In other words, we can approximately recover the parameter $\theta^*$ up to a sign difference: the principal component analysis of $\hat{\Sigma}$ gives either $v$ or $-v$, while $\frac{\theta^*}{\sqrt{d}}$ is close to $v$.

**Lemma 4.** *Under the same setting as Theorem 4, suppose that $n > d$ and $\sigma\sqrt{\frac{\sigma^2+d}{nd}} < \frac{1}{128}$. Then, with probability at least $1 - c_1 e^{-c_2 n \min\{\frac{\sqrt{d}c_3}{\sigma}, (\frac{\sqrt{d}c_3}{\sigma})^2\}}$, there is a unique maximal eigenvector $v$ of the sample covariance matrix $\hat{\Sigma} = \frac{1}{n}\sum_{i=1}^n x_i^U x_i^{U\top}$ with unit $\ell_2$ norm such that*

$$\left\|v - \frac{\theta^*}{\sqrt{d}}\right\|_2 \leq \tau_0 = \min\{c_0 \sigma\sqrt{\frac{\sigma^2+d}{nd}} + c_3, \sqrt{2}\} \tag{16}$$

*Proof.* As discussed above, $\frac{x_i^U}{\sigma}$ is a spiked covariance ensemble. By Theorem 6 we have with probability at least $1 - c_1 e^{-c_2 n \min\{\frac{\sqrt{d}c_3}{\sigma}, (\frac{\sqrt{d}c_3}{\sigma})^2\}}$, there is a unique maximal eigenvector $\tilde{v}$ of the sample covariance matrix $\hat{\Sigma} = \frac{1}{n}\sum_{i=1}^n x_i^U x_i^{U\top}$ such that

$$\left\|\tilde{v} - \frac{\theta^*}{\sqrt{d}}\right\|_2 \leq c_0' \sigma\sqrt{\frac{\sigma^2+d}{nd}} + c_3'. \tag{17}$$

Let $\tau = c_0' \sigma\sqrt{\frac{\sigma^2+d}{nd}} + c_3'$, we have $\left\|\tilde{v} - \frac{\theta^*}{\sqrt{d}}\right\|_2 \leq \tau$. Below we need to consider two cases, $\tau \leq 1$ and $\tau > 1$.

Case 1: $\tau \leq 1$. Let $v = \frac{\tilde{v}}{\|\tilde{v}\|}$, since both $v$ and $\frac{\theta^*}{\sqrt{d}}$ are unit vectors, we have

$$\left\|v - \frac{\theta^*}{\sqrt{d}}\right\|^2 = \|v\|^2 + \left\|\frac{\theta^*}{\sqrt{d}}\right\|^2 - 2\langle v, \frac{\theta^*}{\sqrt{d}}\rangle = 2 - 2\langle v, \frac{\theta^*}{\sqrt{d}}\rangle \tag{18}$$

Recall that $\left\|\tilde{v} - \frac{\theta^*}{\sqrt{d}}\right\|_2 \leq \tau$, which is equivalent to

$$\tau^2 \geq \|\tilde{v}\|^2 + \left\|\frac{\theta^*}{\sqrt{d}}\right\|^2 - 2\langle \tilde{v}, \frac{\theta^*}{\sqrt{d}}\rangle$$

$$= \|\tilde{v}\|^2 + 1 - 2\|\tilde{v}\|\langle v, \frac{\theta^*}{\sqrt{d}}\rangle$$

Rearranging the terms and using AM-GM inequality gives

$$2\langle v, \frac{\theta^*}{\sqrt{d}}\rangle \geq \|v\| + \frac{1-\tau^2}{\|v\|} \geq 2\sqrt{1-\tau^2} \tag{19}$$

Therefore, by equation 18,

$$\left\| v - \frac{\theta^*}{\sqrt{d}} \right\| = \sqrt{2 - 2\langle v, \frac{\theta^*}{\sqrt{d}} \rangle}$$
$$\leq \sqrt{2 - 2\sqrt{1 - \tau^2}}$$
$$= \sqrt{\frac{2\tau^2}{1 + \sqrt{1 - \tau^2}}}$$
$$\leq \sqrt{2}\tau$$
$$= \sqrt{2}(c_0'\sigma\sqrt{\frac{\sigma^2 + d}{nd}} + c_3').$$

By substituting $c_0 = \sqrt{2}c_0'$, $c_2 = \frac{1}{2}c_2'$ and $c_3 = \sqrt{2}c_3'$, we complete the proof.

**Case 2:** $\tau > 1$. Let $v$ be one of $\pm\frac{\tilde{v}}{\|\tilde{v}\|}$ such that the the inner product $\langle v, \theta^* \rangle$ is nonnegative. Since both $v$ and $\frac{\theta^*}{\sqrt{d}}$ are unit vectors, we have

$$\left\| v - \frac{\theta^*}{\sqrt{d}} \right\|^2 = \|v\|^2 + \left\| \frac{\theta^*}{\sqrt{d}} \right\|^2 - 2\langle v, \frac{\theta^*}{\sqrt{d}} \rangle = 2 - \frac{2}{\sqrt{d}}\langle v, \theta^* \rangle \leq 2 \tag{20}$$

Therefore, $\left\| v - \frac{\theta^*}{\sqrt{d}} \right\| \leq \sqrt{2} = \tau_0$. □

Now we have proved that by using the top eigenvector of $\hat{\Sigma}$, we can recover the $\theta^*$ up to a sign difference. Next, we will show that it is possible to determine the sign using the labeled data.

**Lemma 5.** *Under the same setting as Theorem 4, suppose $v \in \mathbb{R}^d$ is a unit vector such that $\left\| v - \frac{\theta^*}{\sqrt{d}} \right\|_2 \leq \tau_0$ where $\tau_0 \leq \sqrt{2}$. Then with probability at least $1 - \exp\left( -\frac{d(1 - \frac{\tau_0^2}{2})^2}{2\sigma^2} \right)$, we have $\text{sign}(y^L \cdot v^\top x^L)v^\top\theta^* > 0$.*

*Proof.* Since $\left\| v - \frac{\theta^*}{\sqrt{d}} \right\|_2 \leq \sqrt{2}$, and both $v$ and $\frac{\theta^*}{\sqrt{d}}$ are unit vectors, we have $v^\top\theta^* > 0$. So the event $\{\text{sign}(y^L \cdot v^\top x^L)v^\top\theta^* \leq 0\}$ is equivalent to the event $\{y^L \cdot v^\top x^L \leq 0\}$, *i.e.*

$$\mathbb{P}[\text{sign}(y^L \cdot v^\top x^L)v^\top\theta^* \leq 0] = \mathbb{P}[y^L \cdot v^\top x^L \leq 0] \tag{21}$$

Recall that $x^L$ is sampled from the Gaussian distribution $\mathcal{N}(y^L \cdot \theta^*, \sigma^2\mathbf{I}_d)$, where $y^L$ is sampled uniformly at random from $\{\pm 1\}$, we have $(y^L x^L)$ follows the Gaussian distribution $\mathcal{N}(\theta^*, \sigma^2\mathbf{I}_d)$. Hence,

$$\mathbb{P}[y^L \cdot v^\top x^L \leq 0] = \mathbb{P}_{(y^L x^L) \sim \mathcal{N}(\theta^*, \sigma^2\mathbf{I}_d)}[v^\top(y^L x^L) \leq 0] = \mathbb{P}_{g \sim \mathcal{N}(0,1)}\left[ g \leq -\frac{\theta^* \cdot v}{\sigma} \right] \tag{22}$$

Moreover, from $\left\| v - \frac{\theta^*}{\sqrt{d}} \right\|_2^2 \leq \tau_0^2$ we can get

$$\langle \theta^*, v \rangle \geq \sqrt{d}(1 - \frac{\tau_0^2}{2}) \tag{23}$$

So, using the Gaussian tail bound $\mathbb{P}_{X \sim \mathcal{N}(0,1)}[X \leq -t] \leq \exp(-t^2)$ for all $t \in \mathbb{R}$, and combining with equation 21, equation 22, equation 23, we have

$$\mathbb{P}[\text{sign}(y^L \cdot v^\top x^L)v^\top\theta^* \leq 0] \leq \exp\left( -\frac{d(1 - \frac{\tau_0^2}{2})^2}{2\sigma^2} \right), \tag{24}$$

as stated in the lemma. □

Armed with Lemma 4 and Lemma 5, we now have a precise estimation of $\theta^*$ in the Gaussian mixture model. Then, we will show that the high precision of the estimation can be translated to low robust risk. To achieve this, we need a lemma from Schmidt et al. (2018), which upper bounds the robust classification risk of a linear classifier $\hat{w}$ in terms of its inner product with $\theta^*$.

**Lemma 6.** *(Lemma 20 in Schmidt et al. (2018)). Under the same setting as in Theorem 4, for any $p \geq 1$ and $\epsilon \geq 0$, and for any unit vector $\hat{w} \in \mathbb{R}^d$ such that $\langle \hat{w}, \theta^\star \rangle \geq \epsilon \|\hat{w}\|_p^*$, where $\|\cdot\|_p^*$ is the dual norm of $\|\cdot\|_p$, the linear classifier $f_{\hat{w}}$ has $\ell_p^\epsilon$ -robust classification risk at most* $\exp\left(-\frac{(\langle \hat{w}, \theta^\star \rangle - \epsilon \|\hat{w}\|_p^*)^2}{2\sigma^2}\right)$.

Lemma 6 guarantees that if we can estimate $\theta^\star$ precisely, we can achieve small robust classification risk. Combine with Lemma 4 and Lemma 5 which provide such estimation, we are now ready to prove the robust classification risk bound stated in Theorem 4. We can actually prove a slightly more general theorem below with some extra parameters, and obtain Theorem 4 as a corollary.

**Theorem 7.** *Let $(x^L, y^L)$ be a labeled data drawn from $(\theta^*, \sigma)$-Gaussian mixture model $\mathcal{P}_{XY}$ with $\|\theta^*\|_2 = \sqrt{d}$. Let $x_1^U, \cdots, x_n^U$ be $n$ unlabeled data drawn from $\mathcal{P}_X$. Let $\tau_0$ be as stated in Lemma 4, and $v \in \mathbb{R}^d$ be the normalized eigenvector (i.e. $\|v\|_2 = 1$) with respect to the maximal eigenvalue of $\sum_{i=1}^n x_i^U x_i^{U\top}$ such that $\left\|v - \frac{\theta^*}{\sqrt{d}}\right\|_2 \leq \tau_0$ with probability at least $1 - c_1 e^{-c_2 n \min\{\frac{\sqrt{d}c_3}{\sigma}, (\frac{\sqrt{d}c_3}{\sigma})^2\}}$. Let $\hat{w} = sign(y^L \cdot v^\top x^L)v$. Then with probability at least $1 - c_1 e^{-c_2 n \min\{\frac{\sqrt{d}c_3}{\sigma}, (\frac{\sqrt{d}c_3}{\sigma})^2\}} - \exp(-\frac{d(1-\frac{\tau_0^2}{2})^2}{2\sigma^2})$, the linear classifier $f_{\hat{w}}$ has $\ell_\infty^\epsilon$-robust classification risk at most $\beta$ when*

$$\epsilon \leq 1 - \frac{\tau_0^2}{2} - \frac{\sigma\sqrt{2\log\frac{1}{\beta}}}{\sqrt{d}}. \tag{25}$$

*Proof.* By the choice of $v$ we have equation 23 holds, *i.e.*

$$\langle \theta^*, v \rangle \geq \sqrt{d}(1 - \frac{\tau_0^2}{2}), \tag{26}$$

with probability at least $1 - c_1 e^{-c_2 n \min\{\frac{\sqrt{d}c_3}{\sigma}, (\frac{\sqrt{d}c_3}{\sigma})^2\}}$.

Applying Lemma 5 to $v$ yields

$$sign(y^L \cdot v^\top x^L)v^\top \theta^* > 0, \tag{27}$$

with probability at least $1 - \exp(-\frac{d(1-\frac{\tau_0^2}{2})^2}{2\sigma^2})$.

Notice that $\hat{w} = sign(y^L \cdot v^\top x^L)v$. So by union bound on events equation 26 and equation 27, we have

$$\langle \theta^*, \hat{w} \rangle = sign(y^L \cdot v^\top x^L)\langle \theta^*, v \rangle \geq \sqrt{d}(1 - \frac{\tau_0^2}{2}), \tag{28}$$

with probability at least $1 - c_1 e^{-c_2 n \min\{\frac{\sqrt{d}c_3}{\sigma}, (\frac{\sqrt{d}c_3}{\sigma})^2\}} - \exp(-\frac{d(1-\frac{\tau_0^2}{2})^2}{2\sigma^2})$.

Since $\|\hat{w}\|_2 = 1$, we have

$$\|\hat{w}\|_\infty^* = \|\hat{w}\|_1 \leq \sqrt{d}. \tag{29}$$

By Lemma 6, we have the $\ell_\infty^\epsilon$-robust error is upper bounded by

$$R_{\mathcal{B}-\text{robust}}(f_{\hat{w}}) \leq \exp\left(-\frac{(\langle \hat{w}, \theta^* \rangle - \epsilon\|\hat{w}\|_\infty^*)^2}{2\sigma^2}\right). \tag{30}$$

Combining this with equation 28, equation 29 and the assumption equation 25, we have

$$\langle \hat{w}, \theta^* \rangle - \epsilon\|\hat{w}\|_\infty^* \geq \sqrt{d}(1 - \frac{\tau_0^2}{2}) - \sqrt{d}\left(1 - \frac{\tau_0^2}{2} - \frac{\sigma\sqrt{2\log\frac{1}{\beta}}}{\sqrt{d}}\right) = \sigma\sqrt{2\log\frac{1}{\beta}}. \tag{31}$$

Hence,

$$R_{\mathcal{B}-\text{robust}}(f_{\hat{w}}) \leq \exp\left(-\frac{\left(\sigma\sqrt{2\log\frac{1}{\beta}}\right)^2}{2\sigma^2}\right) = \beta, \tag{32}$$

with probability at least $1 - c_1 e^{-c_2 n \min\{\frac{c_3}{\sigma}, (\frac{c_3}{\sigma})^2\}} - \exp(-\frac{d(1-\frac{\tau_0^2}{2})^2}{2\sigma^2})$, as stated in the theorem. $\square$

Now we are ready to prove Theorem 4.

*Proof of Theorem 4:* Let $c$ be a constant such that $\sigma \leq c \cdot d^{1/4}$ for sufficiently large $d$. Notice that the $\hat{w}$ in Theorem 4 is same as the $\hat{w}$ in Theorem 7 since the maximal eigenvector of $\sum_{i=1}^n x_i^U x_i^{U\top}$ also maximizes $\sum_{i=1}^n (v^\top x_i^U)^2$ over the unit sphere $\|v\| = 1, v \in \mathbb{R}^d$. Theorem 7 guarantees that with probability at least $1 - c_1 e^{-c_2 n \min\{\frac{\sqrt{d}c_3}{\sigma}, (\frac{\sqrt{d}c_3}{\sigma})^2\}} - \exp(-\frac{d(1-\frac{\tau_0^2}{2})^2}{2\sigma^2})$, $\ell_\infty^\epsilon$-robust classification risk is less then $\beta = 0.01$ for

$$\epsilon \leq 1 - \frac{\tau_0^2}{2} - \frac{\sigma\sqrt{2\log\frac{1}{\beta}}}{\sqrt{d}}$$

$$= 1 - \frac{\tau_0^2}{2} - \frac{c\sqrt{2\log\frac{1}{\beta}}}{d^{1/4}}.$$

Choose $c_3$ to be $\frac{1}{2}$. Since $n = \Omega(d)$, $\frac{1}{2} \leq \tau_0 = c_0\sigma\sqrt{\frac{\sigma^2+d}{nd}} + c_3 \leq \frac{c_4}{d^{1/4}} + \frac{1}{2}$, and consequently $\tau_0^2 \leq \frac{1}{4} + \frac{c_4}{d^{1/4}} + \frac{c_4^2}{\sqrt{d}}$. So by Theorem 7, with probability at least $1 - c_1 \exp\left(-c_2' d^{7/4}\right) - \exp\left(-c\sqrt{d}\right)$, $\ell_\infty^\epsilon$-robust classification risk is less then $\beta = 0.01$ for

$$\epsilon \leq 1 - \frac{\tau_0^2}{2} - \frac{c\sqrt{2\log\frac{1}{\beta}}}{d^{1/4}}$$

$$\leq \frac{3}{4} - \frac{c\sqrt{2\log\frac{1}{\beta}}}{d^{1/4}}.$$

Since $c_4, c$ are numerical constants, there exists a constant $D$ such that when $d \geq D$, $\ell_\infty^\epsilon$-robust classification risk is less then $\beta = 0.01$ for $\epsilon \leq \frac{1}{2}$, thus we have completed the proof. $\square$

