# OpenReview forum: "Adversarially Robust Generalization Just Requires More Unlabeled Data"
_ICLR.cc/2020/Conference — Reject_

### Official Review · AnonReviewer3 · 2019-10-21
**Official Blind Review #3**

**Rating:** 3

**Review:**

The authors study the sample complexity of adversarially robust learning with access to unlabeled samples. Theoretically, they consider the setting of Schmidt et al. 2018 (separating two class-conditional Gaussians) and present an algorithm which can learn a robust classifier with only a few labeled samples and a large number of unlabeled samples (circumventing the sample complexity separation of the original work). Then, the authors propose a modification of the VAT algorithm (Miyato et al. 2018) to train deep networks utilizing unlabeled samples. They find that, empirically, their algorithm achieves better performance compared to standard adversarial training on the labeled samples.

Overall, the paper addresses an interesting problem, studying both a simple theoretical setting and a real-world empirical setting in which the authors achieve an improvement over prior work.

Unfortunately, the paper is concurrent with two other works (which the authors acknowledge: Carmon et al. 2019, Uesato et al. 2019) which have already been accepted for publication at NeurIPS 2019. All of these works are very similar in spirit, proposing an algorithm for the theoretical setting of Schmidt et al. 2018 and an empirical algorithm for real-world settings. Moreover, these works improve over the current manuscript in a number of ways:
-- The algorithm proposed for the theoretical setting is more general and is essentially the same as the algorithm used for real-world dataset.
-- The experimental evaluation is significantly more extensive, performing additional ablations, and exploring the methods in more detail. The work of Uesato et al. 2019 is virtually a superset of the results in this manuscript.
-- Both works collect additional images from an unlabeled and uncurated dataset (Tiny Images) and show that they can utilize them using their proposed approach to improve the state-of-the-art robust accuracy on CIFAR10.

Therefore, given that: a) the results in the current manuscript are essentially a subset of the results appearing in Carmon et al. 2019 and Uesato et al. 2019 and b) these works will have already been published at NeurIPS 2019, 4 months before ICLR 2020, I am afraid I need to recommend rejection.

**Experience Assessment:**

I have published in this field for several years.

**Review Assessment: Checking Correctness Of Derivations And Theory:**

I assessed the sensibility of the derivations and theory.

**Review Assessment: Checking Correctness Of Experiments:**

I carefully checked the experiments.

**Review Assessment: Thoroughness In Paper Reading:**

I read the paper thoroughly.

---

### Official Review · AnonReviewer1 · 2019-10-21
**Official Blind Review #1**

**Rating:** 3

**Review:**

Paper summary: This paper seeks to improve robust generalization performance with the help of unlabeled data. The authors first consider the toy model presented in Schmidt et al. and show how the labeled sample complexity in the robust setting can be lowered to match the standard setting if sufficient unlabeled data is available. They then propose a practical algorithm to improve robust test accuracy and evaluate it on the MNIST and CIFAR datasets.

Comments: The problem the paper seeks to address (bridging the generalization gap in the adversarial setting) is an important one, and the paper is clear and well-written.

As the authors discuss, there have been three (other) independent papers that tackle the same problem (which were accepted at NeurIPS). Even though I tried to evaluate this paper keeping in mind that it was written concurrently, I think it falls short in a couple of important aspects which make it hard to recommend acceptance. In particular:

1. Unlike the other papers, the algorithm discussed in the theoretical section (which is able to reduce sample complexity by leveraging unlabeled data) is entirely different from the one used in practice on MNIST/CIFAR. It would make for a more compelling case if the algorithm used experimentally could also work on the toy model or vice versa (which is the case for Carmon et al. and Uesato et al.).

2 . The empirical evaluation is not detailed enough and there is some inconsistency in the baselines.

- In particular, the authors report that VAT attains poor robustness (<2.5% for both 5k and 10k labeled). However, Uesato et al. also benchmark against VAT in a very similar setting of 4k labelled CIFAR data points (with the same eps=8/255) and get ~32% accuracy (cf. Figure 1 from their paper). I could not find any difference between the two baselines except for the fact that Uesato et al. implement VAT with a KL divergence penalty (as suggested in the VAT paper) instead of cross entropy (as is used in this paper). This is somewhat concerning because based on the baselines reported in Uesato et al., the improvement of the approach proposed in this paper (which comes from doing 7 steps instead of 1 to find the adversarial example) are marginal. (Additionally, in this setting the approach of Uesato et al. gets robustness of about ~45% which is significantly better than ~33% reported in this paper.)

- Moreover, this paper evaluates on much fewer benchmarks (only MNIST/CIFAR with few labeled examples) compared to the other papers (which also study for example SVHN and the impact of using unlabeled ImageNet on CIFAR robustness).

The overlap with concurrent work is unfortunate, and it makes it hard to evaluate this paper. However, my two main concerns are (1) inconsistency in baselines which cast some doubt on the improvements offered by the proposed approach, and (2) the fact that both the algorithm and the experimental evaluation seem to be a subset of that in concurrent work (especially Uesato et al.). Thus, I have to recommend rejection.


**Experience Assessment:**

I have published in this field for several years.

**Review Assessment: Checking Correctness Of Derivations And Theory:**

I assessed the sensibility of the derivations and theory.

**Review Assessment: Checking Correctness Of Experiments:**

I carefully checked the experiments.

**Review Assessment: Thoroughness In Paper Reading:**

I read the paper thoroughly.

---

### Official Review · AnonReviewer2 · 2019-10-24
**Official Blind Review #2**

**Rating:** 3

**Review:**

This paper considers the problem of adversarial robustness. The paper shows that (Theorem 1) robust generalization error can be bounded in terms of the standard generalization error and a stability term, that does not depend on the labels. The paper also shows that for a simple classification problem involving learning the separator for a symmetric 2 gaussian mixture data, we can solve this problem robustly without additional labeled examples. The paper suggests that we can use unlabeled data to improve the robust generalization. Towards this the paper regularizer on the unlabeled data, that promotes stability in the model prediction. The paper evaluates this on Mnist and Cifar showing the better performance of the proposed regularization over PGD adversarial training.

The Theorem 1 in this paper is a triangle inequality on the loss ,and the observation about splitting the robust generalization into standard generalization error and stability, is not particularly new. The earlier work Zhang et al., 2019b show a similar result in their paper. They even propose and experiment with a similar regularizer (see eqs 3 and 5 in Zhang et al., 2019b). The exact implementation while can be different between these two, the paper does not currently compare with this and there is no evidence to prefer this regularizer over the existing one.

The Gaussian setting considered in this paper is quite simple and the techniques developed there are particular to the symmetric 2 Gaussian mixture problem. Given the other parallel works studying the same setting, it is good to also include a comparison of the exact results (such as sample complexity) for this setup.

Overall I find the contributions of this paper to be not sufficient and cannot recommend acceptance at this stage.

Minor:
The last line above theorem 4 and second line after eq 9 are written poorly.

 Zhang et al., 2019b  https://arxiv.org/abs/1901.08573

**Experience Assessment:**

I have read many papers in this area.

**Review Assessment: Checking Correctness Of Derivations And Theory:**

I assessed the sensibility of the derivations and theory.

**Review Assessment: Checking Correctness Of Experiments:**

I assessed the sensibility of the experiments.

**Review Assessment: Thoroughness In Paper Reading:**

I read the paper thoroughly.

---

### Decision · Program_Chairs · 2019-12-19

**Decision:**

Reject

**Comment:**

This work starts with a decomposition of the adversarial risk into two terms: the first is the usual risk, while the second is a stability term, that captures the possible effect of an adversarial perturbation. The insight of this work is that this second term can be dealt with using unlabelled data, which is often in plentiful supply. Unfortunately, the same ideas was developed concurrently and independently by several groups of authors.

The reviewer all agreed that this particular version was not ready for publication. In two cases, the authors compared the work unfavorably with concurrent independent work. I will note that the main bound somewhat ignores the issue of overfitting that the second term deals with via the Rademacher bound. Unless one assumes one has unlimited unlabeled data, could one not get an arbitrarily biased view of robustness from the sample. Seems like a gap to fill.